# The Impact of Different Relative Humidity Levels on the Production Performance, Slaughter Performance, and Meat Quality of White Pekin Ducks Aged 4 to 42 Days

**DOI:** 10.3390/ani13233711

**Published:** 2023-11-30

**Authors:** Dongyue Sun, Congcong Xu, Yi Liu, Zichun Dai, Ziyi Pan, Rong Chen, Rihong Guo, Fang Chen, Zhendan Shi, Shijia Ying

**Affiliations:** 1College of Animal Science and Technology, Nanjing Agricultural University, Nanjing 210095, China; 2021805115@stu.njau.edu.cn; 2Animal Husbandry Research Institute, Jiangsu Academy of Agricultural Sciences, Nanjing 210014, China; 202130321113@bua.edu.cn (C.X.); 2222117009@stmail.ujs.edu.cn (Y.L.); 20210064@jaas.ac.cn (Z.D.); panziyi718@163.com (Z.P.); chenrong_big@163.com (R.C.); rhguo@jaas.ac.cn (R.G.); fchen_m@sina.com (F.C.); zdshi@jaas.ac.cn (Z.S.); 3College of Animal Science and Technology, Beijing University of Agriculture, Beijing 102206, China; 4College of Life Sciences, Jiangsu University, Zhenjiang 212013, China

**Keywords:** Pekin ducks, production performance, meat quality, humidity

## Abstract

**Simple Summary:**

Relative humidity, as a significant environmental factor, has an important effect on the growth of ducks. This study elucidates the impact of different humidity environments on the production performance and muscle quality of ducks. The research findings demonstrate that ducks exhibit better growth and meat quality at 74% humidity; however, when environmental humidity exceeds 81%, production performance and meat quality plummet sharply.

**Abstract:**

This study aimed to investigate the effects of different humidity levels on the growth performance, slaughter performance, and meat quality of Pekin ducks through the artificial control of humidity, and to identify the suitable environmental humidity for Pekin duck growth. A completely randomized single-factor design was employed, selecting 144 newly hatched male Pekin ducks with healthy and similar BW (body weight) (60.92 g ± 4.38). These ducks were randomly assigned to four groups (A (RH (relative humidity) = 60%), B (RH = 67%), C (RH = 74%), D (RH = 81%)), with 12 ducks and 3 replicates in each group. The ducks were raised in a climate-controlled room for 42 days with ad libitum access to feed and water. BW and feed intake were measured every 3 days, and slaughter performance and meat quality were assessed at 42 days. There was no significant difference in the ADG (average daily gain) from 1 to 21 days (*p* > 0.05). The ADFI (average daily feed intake) of Group D was significantly lower than that of Groups A, B, and C (*p* < 0.05), with no significant differences between Groups A, B, and C (*p* > 0.05). At 42 days, the BW, ADG, and ADFI of Groups A and C were significantly higher than those of Group D (*p* < 0.05), with no significant differences among Groups A, B, and C (*p* > 0.05). Group C had a significantly higher breast muscle weight, breast muscle ratio, liver weight, and liver index than Groups B and D (*p* < 0.05), with no significant differences between Groups A, B, and D (*p* > 0.05). The meat shear force in Group C was significantly lower than that in Groups A, B, and D (*p* < 0.05). The L* (brightness) of Group C was significantly lower than that of Group A (*p* < 0.05), and the a* (redness) value of Group C was significantly higher than that of Groups A and B (*p* < 0.05), with no significant difference compared to Group D (*p* > 0.05). Group B had a significantly higher cooking loss than Groups A, C, and D (*p* < 0.05), with no significant differences among Groups A, C, and D (*p* > 0.05). Under 26 °C conditions, Pekin ducks perform best in terms of the production performance and feed efficiency, with high-quality meat, especially when reared at 74% humidity.

## 1. Introduction

China is the world’s leading producer of duck meat. According to statistics from the Food and Agriculture Organization (FAO), China accounts for 85% of the world’s total duck meat production [1,2,3]. Duck meat is favored by consumers due to its high protein and vitamin B content [4,5,6]. Duck feathers are extensively used in the production of jackets and pillows [7]. White Pekin ducks account for 80% of China’s duck meat industry, with an annual average production of 2.9 billion birds. The transition in duck farming practices from traditional water-based systems to intensive dry farming has led to significant changes in the environment, resulting in a considerable decline in animal welfare [8,9], which has had a significant impact on duck production [10]. 

Among the environmental factors affecting duck production, the impact of relative humidity is often overlooked. However, humidity is one of the key factors affecting the duck growth, especially during changes in farming practices. Unsuitable humidity (RH > 80% or RH < 40%) conditions have a negative impact on duck production performance and directly result in reduced economic efficiency in duck farming [11,12,13]. Therefore, it is very important to explore the effects of different relative humidities on the performance and meat quality of Pekin ducks. This is not just good for animal welfare, but also provides valuable insights for the optimization of farming processes and the adjustment of indoor environments in poultry houses.

Environmental factors have a significant effect on livestock production [14]. Relative humidity affects poultry independently of temperature, significantly impacting poultry under suitable temperature conditions [15]. Reduced feed intake is one of the most significant effects of relative humidity on poultry growth, as heat stress affects the length and weight of the digestive tract, thereby affecting feed consumption [16]. Following an increase in relative humidity from 60% to 80%, 42-day-old Pekin ducks exhibited a significant reduction in BW and abdominal fat ratio, along with a significant increase in the FCR (feed conversion ratio). However, the breast muscle ratio, leg muscle ratio, and skin fat ratio remained insignificantly affected [17]. Excessive relative humidity can cause respiratory tract damage in poultry [18,19]. In high-humidity environments, feces and urine generate a significant amount of NH_3_ [20,21], which dissolves in tiny suspended water droplets. This NH_3_ adheres to the respiratory mucosa through poultry respiration, thereby disrupting the respiratory tract and increasing the incidence of poultry tracheitis and bacterial diseases [22,23,24,25,26]. These factors significantly affect production performance and meat quality. Wei et al. observed differences in meat quality in broilers exposed to different relative humidity levels: under 30% relative humidity conditions, there was a decrease in BW, ADFI, and breast muscle L*, with an increase in breast muscle shear force (*p* < 0.05) [27].

In this study, the effects of different relative humidities (humidity range 60–81%) on the production performance, slaughtering performance and meat quality of 4-42-day-old Pekin ducks were investigated using a artificially controlled relative humidity environment.

## 2. Materials and Methods

### 2.1. Experimental Methods

A completely randomized single-factor design was employed, selecting 144 newly hatched male Pekin ducks with healthy and similar BW (60.92 g ± 4.38). These ducks were randomly assigned to 4 groups (A (RH = 60%), B (RH = 67%), C (RH = 74%), D (RH = 81%)), with 12 ducks and 3 replicates in each group. The Pekin ducks were placed in four separate environmental control rooms with free access to feed and water. 

The experiment was started when the ducks were 4 days old. From days 4 to 14, the temperature gradually decreased from 35 °C to 26 °C, and from days 15 to 42, the temperature remained at 26 °C (Table 1), light was provided for 20 h from 5 a.m., and the room was dark for the rest of the day, with all other conditions being consistent across the treatment groups, except for relative humidity. Ducks were immunized according to the regulations of the Ministry of Agriculture (Decree Agriculture and animal husbandry [2021] No. 2 of Ministry of Agriculture China on 11 January 2021). BW and feed intake were measured every 3 days, and at 42 days of age, slaughter performance and muscle quality were assessed.

The experimental procedures were approved by the Research Committee of Jiangsu Academy of Agricultural Sciences and conducted in adherence to the Regulations for the Administration of Affairs Concerning Experimental Animals (Decree No. 63 of the Jiangsu Academy of Agricultural Science on 8 July 2014).

### 2.2. Experimental Management

The experiment was conducted in the artificial climate-controlled rooms at the Poultry Farm of the Luhe Animal Science Base of Jiangsu Academy of Agricultural Sciences in Jiangsu, China, from March to May 2022. The temperature and relative humidity for different age groups of ducks were automatically adjusted according to the preset parameters (Table 1) with a temperature accuracy of ±1 °C and a relative humidity accuracy of ±5%. All experimental ducks were reared in 4 identical enclosed chambers with dimensions of 4 m × 2.5 m × 2.6 m. The walls and floors of the chambers were constructed using insulation materials. Temperature and relative humidity settings during the experiment were referenced from Table 1. All ducks were raised on polyethylene prefabricated leakage dung plates. The leakage dung plates were 60 cm above the ground and fixed with galvanized steel frames. Dry rice husks, 20 cm thick, were spread on the ground. Temperature and relative humidity sensors were fixed 60 cm above the leakage dung plates to monitor the indoor temperature and relative humidity in real time.

During the experiment, ducks had free access to food and water. The water was provided through nipple drinkers, and the height was adjusted periodically according to the ducks’ growth. The experimental diet consisted of standard commercial feed, meeting the hygiene standards of GB13078-2017 [28], and the nutritional levels complied with the requirements of the NY/T2122—2012 [29] standard for meat duck feeding. The composition of the feed is detailed in Table 2. Continuous light was provided for 24 h a day from day 1 to day 3, and during days 4 to 42, a light–dark cycle of 20 h of light and 4 h of dark was implemented.

### 2.3. Duck and Slaughter Performance

Feed intake was recorded, and every 3 days, all ducks were weighed. At 42 days, the empty stomach weight of ducks was measured to calculate the production performance. 

At 42 days, 9 ducks (3 in each replicate) were randomly selected from each group and slaughtered after weighing the fasting body weight, stripping the liver, spleen, abdominal fat, pectoral muscle, leg muscle and abdominal fat, with the relevant ratios subsequently calculated.
Liver index = (liver weight/fasting body weight) × 100%
Spleen index = (spleen weight/fasting body weight) × 100%
Abdominal fat index = (abdominal fat weight/fasting body weight) × 100%
Breast Muscle Ratio (%) = (breast muscle weight/fasting body weight) × 100%
Leg Muscle Ratio (%) = (leg muscle weight/fasting body weight) × 100%
Abdominal Fat Ratio (%) = (abdominal fat weight/fasting body weight) × 100%

### 2.4. Meat Quality and Myofiber 

The pH value, meat color, shear force, and cooking loss indexes were determined using the pectoral muscle obtained from slaughter.

Measurement of pH Value: The breast muscle obtained from the slaughter was used as the material for measuring the pH value. Three fixed positions were selected on the breast muscle of the experimental ducks. The pH value of each position was determined using the portable meat acidity meter (testo 206-pH3, Testo SE & Co. KGaA, Lenzkirch, Germany), and the average value was used to represent the pH measurement of the breast muscle. 

Meat Color Measurement: The breast muscle from one side of the experimental ducks was separated, and three fixed positions were chosen on each piece of breast muscle to measure meat color L* (brightness), a* (redness), and b* (yellowness) values. Each of these measurements was repeated three times using a colorimeter (CR-400, Konica Minolta Sensing, lnc., Tokyo, Japan)

Warner–Bratzler Shear Force Measurement: A texture analyzer (C-LM3B, Northeast Agricultural University, Haerbin, China) was used to determine the shear force of the cooked breast and thigh samples. At least two subsamples of 2 cm × 3 cm × 0.5 cm were cut parallel to the muscle fibers. The crosshead speed was set to 20 cm/min, and the shear force was calculated following the method described by Wattanachant et al. [30].

Cooking Loss: A sample of breast muscle from all experimental ducks was taken, weighed, sealed, and maintained at a central temperature of 70 °C. After cooking in a water bath for 15 min and cooling to room temperature, it was weighed again. The cooking loss was calculated using the following formula: (initial weight–final weight) divided by initial weight, multiplied by 100%.

Protein Content in Muscle: Protein content in the muscle was determined according to the GB 5009.5-2016 [31] standard using an automatic Kjeldahl nitrogen analyzer (VAP500C, Gerhardt, Königswinte, Germany).

Total Ash Content: After weighing the sample, the sample was fully charred to a smoke-free state by heating with a low flame, and then placed in a muffle furnace (QDSH-7, YI−HENG, Shanghai, China) and burned at 550 ± 25 °C for 4 h. The mixture was cooled to about 200 °C, removed, and cooled in a desiccator for 30 min. The burn was repeated to a constant weight, and the ash content was calculated, following the GB 5009.4-2010 [32] standard.

Crude fat was determined using the Soxhlete extraction method according to GB/T5009.6-2003 [33]. The instrument used was a crude fat measuring instrument (S2F-06A, Xinjia, Shanghai, China).

Muscle Fiber Morphological Analysis: During the slaughter, a 5 mm × 5 mm × 15 mm section of breast muscle tissue was taken along the direction of the muscle fibers and fixed in 4% paraformaldehyde universal tissue fixative for 24 h, dehydrated, and embedded in paraffin wax. Tissue sections (3 µm) were stained with hematoxylin and eosin (H&E). The changes in muscle morphology were visualized using a light microscope (ECLIPSE Ti, Nikon, Tokyo, Japan) and the NIS-Elements 4.0 software. The ImageJ 2.0.0 software was used for muscle fiber diameter analysis.

### 2.5. Statistical Analysis

The experimental data were analyzed using SPSS 26 software through one-way analysis of variance (ANOVA). Multiple comparisons were conducted using Duncan’s multiple range test. A significance level of *p* < 0.05 was used to determine significant differences. The data were presented in the format of “mean ± standard deviation”.

## 3. Results

### 3.1. Environmental Parameter Monitoring

Figure 1 is plotted based on data transmitted by the temperature and relative humidity sensors. The results indicate that the experimental conditions conform to the requirements of the experimental design.

### 3.2. Growth Performance

In the early growth phase of Pekin ducks (1–21 days), there were no significant differences in ADG (*p* > 0.05). However, Group D exhibited significantly lower ADFI than Groups A, B, and C (*p* < 0.05), with no significant differences among Groups A, B, and C (*p* > 0.05). The FCR showed a stepwise decrease from Groups A to D. Group A had a significantly higher FCR than Groups C and D (*p* < 0.05), and Groups B and C had significantly higher ratios than Group D (*p* < 0.05), and no significant differences were observed among Groups A, B, and C (*p* > 0.05) (Figure 2).

In the later growth phase of Pekin ducks (22–42 days), Groups A and C displayed significantly higher ADG than Group D (*p* < 0.05), with no significant differences among Groups A, B, and C (*p* > 0.05), and no significant differences between Groups B and D (*p* > 0.05). Average daily feed intake in Group C was significantly higher than in Groups A, B, and D (*p* < 0.05), with no significant differences among Groups A, B, and D (*p* > 0.05) (Figure 2).

Throughout the entire growth period of Pekin ducks (1–42 days), the differences in ADG were similar to the later growth phase. Groups A and C had significantly higher daily weight gain than Group D (*p* < 0.05), with no significant differences among Groups A, B, and C (*p* > 0.05). Average daily feed intake in Groups A and C was significantly higher than in Group D (*p* < 0.05), with no significant differences among Groups A, B, and C (*p* > 0.05). The FCR was lowest in Group C and highest in Group D, although there were no significant differences between the groups (*p* > 0.05) (Figure 2).

The weight gain trend of ducks in the four experimental groups was almost the same. There was no separation of weight differences before 22 days of age. After 22 days of age, the weight advantage of group C gradually appeared (Figure 3a–c). Feed intake also reflects differences in period (Figure 3e).

### 3.3. Slaughter Performance

According to Table 3, Group C exhibited significantly higher liver weight and liver index compared to Groups A, B, and D (*p* < 0.05), with no significant differences observed among Groups A, B, and D (*p* > 0.05). Group C displayed a significant increase in breast muscle weight compared to Groups B and D (*p* < 0.05), with no significant difference compared to Group A, and no significant differences among Groups A, B, and D (*p* > 0.05). Group C demonstrated a significant increase in breast muscle percentage compared to Groups A, B, and D (*p* < 0.05), with no significant differences among Groups A, B, and D (*p* > 0.05). 

### 3.4. Meat Quality and Muscle Microscopic Characteristics

Compared to Group C, the muscle shear force in Groups A, B, and D is significantly reduced (*p* < 0.05), and there are no significant differences among Groups A, B, and D (*p* > 0.05). The L* values in Group C were significantly lower than those in Group A (*p* < 0.05), and no significant differences were observed among Groups B and D (*p* > 0.05). In comparison to Group C, both Groups A and B demonstrate significantly reduced a* values (*p* < 0.05), with no significant differences in Group D (*p* > 0.05); when compared to Group A, Groups C and D display a significant increase in a* (*p* < 0.05), and there are no significant differences in a* in Group B (*p* > 0.05). The cooking loss of Group B was significantly higher than that of Groups A, C, and D (*p* < 0.05), and there was no significant difference among Groups A, C, and D (*p* > 0.05). The cross-sectional area of muscle fibers in Group D was significantly higher than that in Group A, B, and C (*p* < 0.05), and no significant differences were observed among Groups A, B, and C (*p* > 0.05) (Table 4). By looking at the Figure 4, clear group D muscle cross-sectional area more coarse than other patients. There are no significant differences in pH value, b*, fat, protein, ash content, and muscle fiber density (*p* > 0.05) (Table 4).

## 4. Discussion

Housing relative humidity is one of the important factors that affects the growth and development of White Pekin ducks [34,35,36]. According to the test results, relative humidity significantly affects the final weight of White Pekin ducks at the same environmental temperature. Excessively high humidity can disrupt the thermal balance of ducks. When the relative humidity in the air exceeds 80%, the evaporative cooling pathway of White Pekin ducks is greatly inhibited, breaking their original heat production–dissipation balance and reducing their ventilation frequency, leading to heat stress, which significantly affects the productivity of White Pekin ducks. Group C White Pekin ducks had the highest BW, while Group D White Pekin ducks had the lowest BW, and the BW of Groups A and B fell between those of Groups C and D. This suggests that there may be an optimal relative humidity level for White Pekin duck growth, with growth being unfavorable when it is above or below this peak, which is consistent with the research of Sun et al. [37]. Sun et al. found that AA broilers (the Arbor Acres broiler, also known as the AA broiler, is a four-line matching white-feathered broiler breed bred by Arbor Acres Breeding Company in the United States) subjected to 30% and 90% humidity treatments had significantly lower market weight compared to AA broilers subjected to 60% humidity treatment, which aligns with the results of this experiment. It has also been found that appropriately increasing relative humidity can make the skin of chickens’ legs and feet shinier and increase their market weight [38,39]. In Wei et al.’s research, AA broilers grown for 42 days at relative humidity level A had a higher market weight, daily weight gain, and daily feed intake than broilers grown in high humidity (85%) and low humidity (35%) environments [27,40]. Yahav et al. found that the feed conversion rate of poultry is almost unaffected by changes in relative humidity [41], which is consistent with the results of this experiment. However, Wei et al.’s research negated this point, as AA broilers grown at relative humidity level A had a significantly higher feed conversion rate than those subjected to 35% humidity and 85% humidity treatments. The reason for this may be the different breeds of experimental animals used in the two experiments, or it may be due to differences in the relative humidity range set in the experimental design. The relative humidity range in Yahav et al.’s experiment was 40% to 75%, divided into three levels, which is relatively close to the relative humidity range set in this experiment (60% to 81%), and the conclusions reached are consistent with this experiment. In Wei et al.’s experiment, the relative humidity range spanned a wider range, and Wei et al. and others used AA broilers, while Yahav et al. used Cobb broilers, which may result in some deviations in the experimental results. 

Slaughter performance is influenced by various factors, such as the environmental temperature and relative humidity, farming methods, stocking density, and feed nutritional levels [42,43,44]. It is an important indicator for measuring poultry production performance and economic efficiency. However, there have been no reports regarding the impact of relative humidity on the slaughter performance of White Pekin ducks. Slaughter performance is highly correlated with production performance, and muscle development and fat metabolism are closely related to the growth and activity level of animals. 

The ratio of pectoralis muscle in Group C was significantly higher than that in the other groups (*p* < 0.05). This means that ducks grown in a 74% humidity environment can produce more chest muscle, which is good news for producers and consumers. However, relative humidity changes did not seem to have a significant effect on leg muscle mass. Even so, Group C had the highest combined muscle percentage (pectoral muscle percentage + leg muscle percentage). We speculate that the cause of this phenomenon is related to feed intake, which is first affected by humidity changes and then leads to muscle growth and development. Another possible reason is that humidity affects the respiration rate of the ducks. Studies have shown that humidity affects the respiratory rate of poultry [45]. An increased respiratory rate leads to more frequent use of the pectoralis muscle, but whether this leads to a significant increase in the growth rate of the pectoralis muscle has not been tested so far. In addition, there is still a big gap between the experimental site and the standardized breeding farm, and the duck activity site is relatively small, which may also be one of the reasons for the lack of difference in the leg muscle growth rate. In this experiment, there was no significant difference in abdominal fat in White Pekin ducks (*p* > 0.05). However, research by Sun et al. on broilers showed that both excessively low and excessively high relative humidity can reduce the abdominal fat percentage of broilers at 21 days of age [39]. The reason for this difference may be due to variations in the experimental animals used, differences in abdominal fat deposition between 21 days and 42 days of age, and the possibility that the impact of humidity on abdominal fat deposition varies between White Pekin ducks and broilers.

Adequate development of the internal organs is fundamental for the efficient deposition of nutrients in muscle and bone tissues. The digestion and absorption of nutrients can be directly affected by the growth and development of internal organs. The organ index is one of the important indicators reflecting the growth and development of organs [46]. There are many studies showing that temperature, diet composition and other factors can have significant effects on organ [47,48,49] indices. However, no report on the effect of relative humidity on the duck organ index has been found so far. The spleen is an important immune organ and the largest peripheral lymphoid organ in birds [50]. We generally assume that a higher spleen index represents a higher immune capacity. Apparently, the change in humidity did not have an excessive effect on the spleen, or the effect was not yet apparent. However, 74% relative humidity increased the duck’s liver index. Liver is an important organ that supplies endogenous fat to the body and plays an important role in the process of lipid digestion and metabolism. This suggests that relative humidity affects nutrient metabolic processes in ducks.

One of the primary objectives of animal husbandry is to provide high-quality protein sources for humans. Moreover, in many regions around the world, poultry has consistently served as the primary source of meat [51]. In practical production, the quality of meat is assessed based on multiple indicators, including the pH value, shear force, cooking loss, and meat color [52].

The pH value is related to the glycogen content in the muscles at the time of slaughter and post-slaughter glycolysis [53]. In this study, there was no significant change in pH values, indicating that 60–81% humidity does not affect the post-slaughter glycolysis rate in Pekin duck muscles. Shear force is a critical parameter for assessing muscle tenderness. This measure has a direct impact on the texture and flavor of meat products, where easier fiber shearing leads to reduced chewiness. This is an important characteristic in muscle quality that is significant for both consumers and producers [54]. Tenderness is collectively determined by several factors, including the thickness of muscle fibers, the number of muscle fibers per unit area on the longitudinal vertical cross-section in the direction of muscle filaments, and intermuscular fat. The content of intermuscular fat has been a particular focus for breeders [55]. A study on the meat quality of different types of meat ducks revealed that slow-growing ducks tend to have lower shear force; the reason for this may be the higher protein and fat content in their muscles [56]. In this study, Group C exhibited significantly lower shear force compared to the other treatment groups, although there were no significant changes in protein and fat. The reason for this result may be the variations in the meat duck breeds and rearing time. Furthermore, the size of the cross-sectional area of muscle fibers is commonly considered a significant factor contributing to differences in shear force. Higher meat quality is associated with a smaller cross-sectional area [57]. This was confirmed in the current experiment. Some opinions propose that the cross-sectional area of fibers is associated with the type of muscle fibers, and the type of muscle fibers in poultry may change with exercise levels during growth [58,59,60]. However, this experiment did not test this viewpoint. Additionally, there is evidence of a negative correlation between shear force, the cross-sectional area of muscle fibers, and cooking loss, meaning that muscles with a smaller fiber size tend to have higher cooking losses [61]. This was also demonstrated in this experiment.

Relative humidity is, at all temperatures and pressures, defined as the ratio of the water vapor pressure to the saturation water vapor pressure (over water) at the gas temperature [62]. The definition of relative humidity makes this environmental parameter difficult to measure and control because it is a ratio rather than an absolute value. Under different temperature conditions, the relative humidity values are different, even if the value of water vapor content in the air is the same.

The control logic of the environmental control room in this test is to first allow the ambient temperature reach the set parameters, and then the relative humidity is adjusted. This causes the humidity regulation to be relatively lagged.

When the relative humidity is below the set value, the water is atomized by ultrasonic waves and then blown into the room; when the relative humidity is higher than the set value, the air in the room is pumped out for dehumidification and returned to the room. The relative humidity of outdoor air varies greatly during the day and night, and this air is blown into the room via the fresh air system without the relative humidity being changed. In the environmental control chamber, the breathing of the ducks, the dripping of the drinking nipple, and the discharge of feces changed the content of water vapor in the air. Therefore, under the influence of all relevant factors, the relative humidity fluctuated continuously within a certain range, forming a state of dynamic equilibrium. This can be proved by our measured data (Figure 1). In order to minimize the fluctuation amplitude, we can increase the adjustment capacity of the environmental control chamber, such as using more powerful machines. However, we were not able to achieve complete humidity stabilization, as this is extremely difficult to achieve based current technology. This may have a small effect on the accuracy of the test results, and is unavoidable. Nevertheless, the results obtained in this trial are very relevant.

## 5. Conclusions

Relative humidity significantly affected growth performance, slaughter performance, and meat quality of White Pekin ducks aged from 4 to 42 days. These effects were mainly reflected in the body weight, feed intake, pectoralis muscle rate, liver index, muscle shear force, meat color and muscle morphology. Relative humidity had little effect on Pekin ducks when it was between 60% and 74%. However, when relative humidity reached 81%, the body weight and muscle quality of the ducks decreased significantly. There may be a suitable humidity range for the growth of Pekin ducks, rather than a fixed value. The upper limit, lower limit and peak of this range are worth exploring further. At present, based on all evaluation indicators, the growth performance and meat quality of White Pekin duck are the best under the condition of 74% humidity. Pending further validation, this can be used to guide livestock production.

## Figures and Tables

**Figure 1 animals-13-03711-f001:**
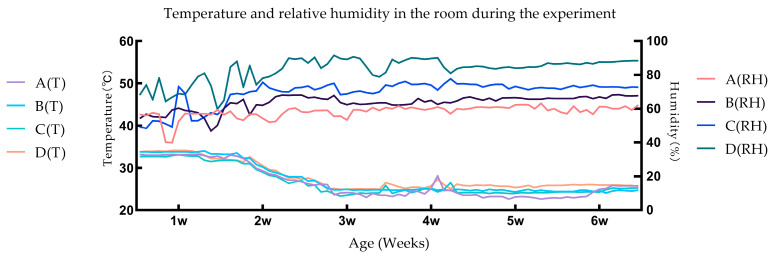
Temperature and relative humidity in the room during the experiment.

**Figure 2 animals-13-03711-f002:**
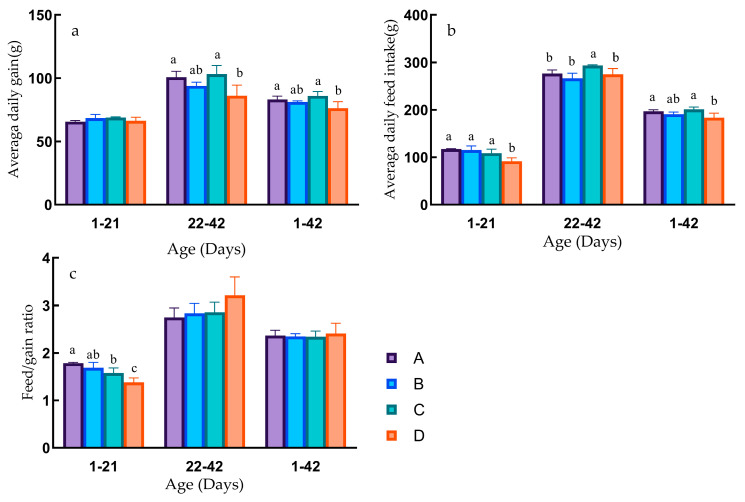
Different growth stages production performance. Different lowercase letters indicate significant differences (*p* < 0.05). The same letter or no letter indicates that the differences are not significant (*p* > 0.05). (**a**) Average daily gain at different growth stages. (**b**) Average daily feed intake at different growth stages. (**c**) Feed conversion rate at different growth stages.

**Figure 3 animals-13-03711-f003:**
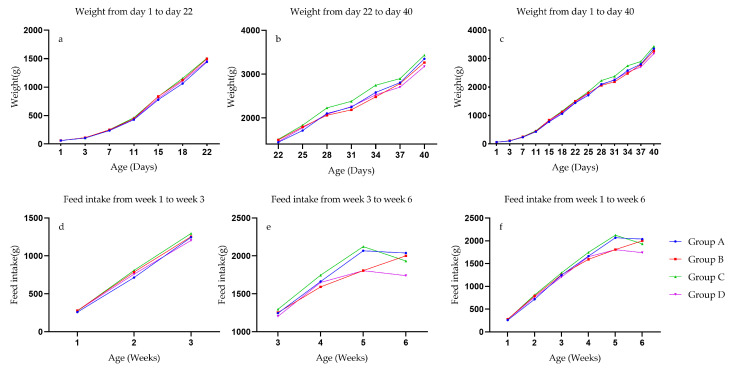
Body weight trend and feed intake accumulation of Pekin ducks. (**a**) Four patients 1 to 22 days duck body weight changes. (**b**) Four patients 22 to 40 days duck body weight changes. (**c**) Four patients 1 to 40 days duck body weight changes. (**d**) Four patients 1 to 3 weeks ducks’ feed intake changes. (**e**) Four patients 3 to 6 weeks ducks’ feed intake changes. (**f**) Four patients 1 to 6 weeks ducks’ feed intake changes.

**Figure 4 animals-13-03711-f004:**
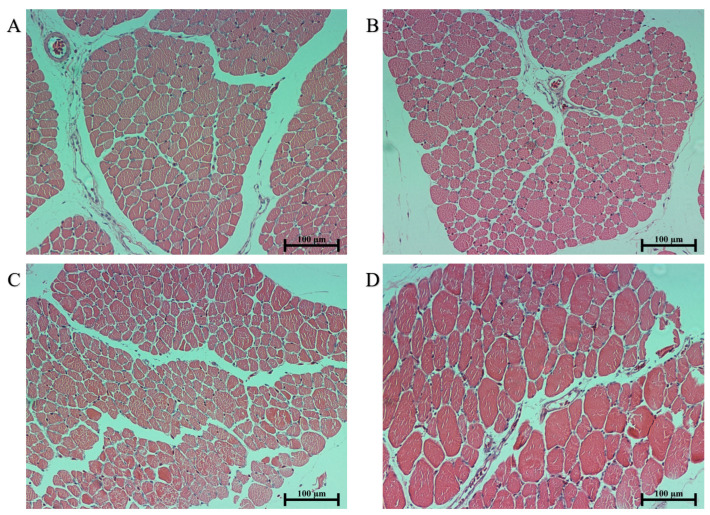
Histological sections of pekin duck pectoral muscles in different groups at 42 days of age (hematoxylin and eosin staining. Scale bar = 100 μm). (**A**) Histological sections of the pectoral muscles from group A. (**B**) Histological sections of the pectoral muscles from group B. (**C**) Histological sections of the pectoral muscles from group C. (**D**) Histological sections of the pectoral muscles from group D.

**Table 1 animals-13-03711-t001:** Environmental parameter setting for 1–42 days.

Day	Temperature All Groups Are Identical	Relative Humidity			
		A	B	C	D
1–3	35 °C	60%			
4	35 °C	60%	67%	74%	81%
5	34 °C	60%	67%	74%	81%
6	34 °C	60%	67%	74%	81%
7	34 °C	60%	67%	74%	81%
8	33 °C	60%	67%	74%	81%
9	32 °C	60%	67%	74%	81%
10	31 °C	60%	67%	74%	81%
11	30 °C	60%	67%	74%	81%
12	29 °C	60%	67%	74%	81%
13	28 °C	60%	67%	74%	81%
14	27 °C	60%	67%	74%	81%
15–42	26 °C	60%	67%	74%	81%

This table shows the temperature and relative humidity that each treatment group should reach at different times.

**Table 2 animals-13-03711-t002:** Composition and nutrient levels of basal diet (%, air-dry basis) in 1–14 days and 15–42 days.

Ingredient	Dietary Energy (1–14 Days)	Dietary Energy (15–42 Days)	Calculated Nutrition Levels	Dietary Energy (1–14 Days)	Dietary Energy (15–42 Days)
Corn	62.10	68.5	ME (Kcal/kg)	2900	2950
Wheat	5.22	3.02	Crude protein (%)	20.01	17.52
Soybean meal	28.60	24.30	Ca (%)	0.90	0.85
Rapeseed meal	0.00	0.60	Available P (%)	0.42	0.40
Limestone	0.93	0.90	Digestible lysine (%)	0.98	0.82
CaHPO4 (2H2O)	1.85	1.72	Digestible methionine (%)	0.47	0.37
NaCl	0.34	0.33	Digestible Met + Cys (%)	0.75	0.64
Choline chloride	0.15	0.15	Digestible threonine (%)	0.65	0.56
Premix *	0.23	0.23	Digestible tryptophan (%)	0.28	0.19
*DL*-methionine	0.24	0.12			
*L*-lysine·HCl	0.20	0.10			
*L*-tryptophan	0.08	0.02			
*L*-threonine	0.06	0.01			
Total	100	100			

* Supplied per kilogram diet: vitamin A, 9,000 IU; vitamin D3, 2,000 IU; vitamin E, 10 IU; vitamin B1, 2 mg; vitamin B2, 4.8 mg; pantothenic acid, 20 mg; vitamin B12, 0.02 mg; folic acid, 1 mg; niacin 50 mg; Cu (CuSO4·5H2O), 8 mg; Fe (FeSO4·7H2O), 60 mg; Zn (ZnSO4·7H2O), 60 mg; Se (NaSeO3), 0.3 mg; I (KI), 0.4 mg.

**Table 3 animals-13-03711-t003:** Slaughter performance.

Item	Grouping				*p*
	A	B	C	D	
Liver (g)	74.2 ± 2.70 ^b^	73.5 ± 4.38 ^b^	89.44 ± 6.84 ^a^	70.03 ± 3.22 ^b^	0.025
Liver index	2.09 ± 0.02 ^b^	2.11 ± 0.07 ^b^	2.42 ± 0.08 ^a^	2.14 ± 0.00 ^b^	0.040
Spleen (g)	2.81 ± 0.18	2.94 ± 0.29	3.18 ± 0.15	2.88 ± 0.12	0.570
Spleen index	0.08 ± 0.00	0.09 ± 0.01	0.09 ± 0.00	0.09 ± 0.00	0.440
Abdominal fat (g)	36.72 ± 2.33	34.01 ± 3.95	38.16 ± 0.72	34.99 ± 3.89	0.080
Abdominal fat index	1.03 ± 0.06	0.98 ± 0.11	1.04 ± 0.04	1.06 ± 0.08	0.170
Breast Muscle (g)	421.87 ± 12.71 ^ab^	405.67 ± 19.14 ^b^	478.73 ± 17.2 ^a^	384.54 ± 28 ^b^	0.015
Breast Muscle Ratio (%)	11.85 ± 0.31 ^b^	11.66 ± 0.34 ^b^	13.04 ± 0.29 ^a^	11.67 ± 0.53 ^b^	0.040
Leg Muscle (g)	324.66 ± 8.95	312.88 ± 12.71	327.56 ± 14.02	299.88 ± 7.30	0.292
Leg Muscle Ratio (%)	9.12 ± 0.22	9.02 ± 0.29	8.91 ± 0.19	9.22 ± 0.22	0.855

Different lowercase letters indicate significant differences (*p* < 0.05). The same letter or no letter indicates that the differences are not significant (*p* > 0.05).

**Table 4 animals-13-03711-t004:** Meat quality and muscle microscopic characteristics.

Item	Grouping				*p*
	A	B	C	D	
pH value	6.45 ± 0.09	6.53 ± 0.03	6.38 ± 0.09	6.41 ± 0.07	0.581
Shear force (N)	65.39 ± 2.31 ^a^	69.71 ± 2.32 ^a^	58.82 ± 4.86 ^b^	65.4 ± 2.58 ^a^	0.033
Brightness (L*)	35.24 ± 0.63 ^a^	34.92 ± 0.55 ^ab^	33.01 ± 0.43 ^b^	34.4 ± 0.86 ^ab^	0.044
Redness (a*)	15.81 ± 0.61 ^c^	16.68 ± 0.53 ^bc^	18.28 ± 0.24 ^a^	17.48 ± 0.47 ^ab^	0.006
Yellowness (b*)	1.56 ± 0.31	1.87 ± 0.38	1.23 ± 0.19	1.73 ± 0.29	0.383
Fat	1.256 ± 0.191	1.533 ± 0.314	1.178 ± 0.146	1.689 ± 0.492	0.638
Protein	20.978 ± 0.375	21.167 ± 0.534	22.267 ± 0.233	22.167 ± 0.481	0.077
Ash content	1.911 ± 0.152	1.511 ± 0.115	1.744 ± 0.151	1.789 ± 0.125	0.234
Cooking loss (%)	20.61 ± 1.07 ^b^	24.09 ± 0.76 ^a^	20.54 ± 1.29 ^b^	18.62 ± 1.31 ^b^	0.014
Muscle fiber density (N/mm²)	1565.6 ± 180.97	1473.34 ± 66.16	1545.01 ± 146.71	1090.15 ± 241.82	0.101
Muscle fiber cross-sectional area (CA/μm²)	560.93 ± 109.87 ^b^	532.66 ± 78.85 ^b^	608.88 ± 83.43 ^b^	1352.41 ± 379.05 ^a^	0.030

Different lowercase letters indicate significant differences (*p* < 0.05). The same letter or no letter indicates that the differences are not significant (*p* > 0.05).

## Data Availability

The data that support the findings of this study are available from the corresponding author upon reasonable request.

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
