# Peer review of "The Impact of Different Relative Humidity Levels on the Production Performance, Slaughter Performance, and Meat Quality of White Pekin Ducks Aged 4 to 42 Days"

_animals, 2023, doi:10.3390/ani13233711_

Round 1

Reviewer 1 Report

Comments and Suggestions for Authors

Within the text provided, there are numerous grammatical errors particularly with the confusion of tenses, which significantly impact its clarity and coherence. There is a substantial need for improvement in terms of writing and structure. It is essential to address these issues to enhance the formal tone and overall readability of the document, ensuring that it meets the hight standards expected of academic writing. Additionally, careful attention to syntactic precision and consistency in terminology will further refine the text´s quality.

Paper review:

“The Impact of Different Humidity Levels on the Production Performance, Slaughter Performance, and Meat Quality of White Pekin Ducks Aged 4 to 42 Days.”

 Line 12: Consider rewriting the sentence “Humidity is a crucial environmental factor influencing duck growth”. For instance, “Humidity plays a vital role as an environmental factor significantly impacting the growth of ducks”.

 Line 14: you could put “the research findings demonstrate” instead of reveal.

Line 15: maybe you could name types of “relevant indicators”.

 Line 19: it is suggested to write the number one hundred forty-four with digits (144).

 Line 20: What does “good spirits” mean? consider other options.

 Line 21: you already put randomized design. It is suggested to delete the word “randomly”.

 Line 21: Write the meaning of “RH”.

 Line 33: write the meaning of L* and a*.

 Line 42: Consider rewriting the sentence “China is the world`s leading producer of duck meat, and according…” to “China is the world`s leading producer of duck meat. According to statistics…”

 Line 44: the concept “Highly nutritious” is not clear.

 Line 45: replace “, and its feathers” to “. Their feathers”

Line 52: replace “affecting” to “that affect the”

Line 53: maybe you could define unsuitable humidity conditions.

Line 54: delete “Investigating the”

Line 57: consider rewriting the sentence to “ it provides valuable insights for the optimization of farming processes and the adjustment of indoor environments in poultry houses”

Line 73: here you put Wei et al. but in line 231 you put Sun Yongbo and others. (standardize the concepts)

Line 79: controlled instead controlling.

Line 83: define good mental status.

Line 85: rewrite to “with 12 ducks, and three replicates in each group”

Line 90: what are the guidelines? Define them.

Line 97: what are the preset parameters? Include the citation.

Line 111: rewrite “and calculating the relevant ratios” to “with the relevant ratios subsequently calculated.”

Line 119: I don’t understand if it is a title or what. Improve the writing.

Line 157: you must put a title for the figure.

Line 160: delete the “-“ in exper-imental.

Line 220: change “affecting” to “that affects”.

Line 232: explain what AA broilers means.

Introduction

The introduction effectively establishes the importance of duck meat production in China and the predominance of White Pekin duck in the industry.

Technical terms such as “relative humidity”, “production performance”, and “meat quality” are used accurately and consistently.

The term “meat to feed ratio” might benefit from a brief explanation or definition upon first use to aid readers unfamiliar with the term.

Materials and methods

Specific details about the feed given to the ducks (Composition, brand, etc) could be provided for complete transparency and reproducibility.

It is important to mention any ethical consideration or approvals obtained for working with live animals, in accordance with the journal`s requirements and international standards.

Results

The section could benefit from a more detailed information about the non-significant results to provide a complete picture of the data.

It would be useful to include a brief interpretation of the findings alongside the presentation of the data to help readers understand the implications of the results.

Discussion

A discussion on the potential limitations of the experimental setup regarding the results should be considered.

It would be beneficial if the discussion section also addressed any limitations of the study to provide a balanced view and suggest directions for future research.

Conclusion

It succinctly encapsulates the key results, providing a clear takeaway for the reader.

Conclusion could potentially draw more direct connections to practical applications in duck farming and suggest specific changes to current practices based on the study`s findings.

Comments on the Quality of English Language

A comprehensive revieq is required  to improve the quality of English

Author Response

Thank you for your comments. For replies to comments, please refer to the attachment. Thank you again for your interest and support in this article.

Reviewer 2 Report

Comments and Suggestions for Authors

Comments to the Author/s

LN: Line Number

Title: The Impact of Different Humidity Levels on the Production Performance, Slaughter Performance, and Meat Quality of White Pekin Ducks Aged 4 to 42 Days

Title:

LN4: Please delete the full stop sign. In the text it says parameters were taken from day-old. This should be concerned in the title. 

Abstract:

LN 20: Please mention the sex of the birds.

LN20: Please provide BW ± SD in parenthesis.

LN25: Please indicate the P-value in parenthesis (P>0.05)

LN27: Please indicate the P-value in parenthesis (P>0.05)

LN29: Please indicate the P-value in parenthesis (P>0.05)

LN32: Please indicate the P-value in parenthesis (P>0.05)

LN36: Please indicate the P-value in parenthesis (P>0.05)

Introduction:

LN69: Please subscript 3 as: NH3

LN75 & 76: Please remove P- values present.

LN77-79: The authors should be able to highlight the research gap and should be able to describe it in detail. E.g. Limitation of values pertaining to production and meat quality parameters of Pekin ducks aged between 4-42-d.

Methodology:

LN83: Please provide the details on sex of the birds used and BW ±SD values.

LN87: 1-42d is contradictory with what appears in the title (4-42-d)

LN88: Please provide the lighting schedule applied during the research.

LN123: Please provide the manufacturer details and a reference.

LN127: Please provide the model number and manufacturer details of the colorimeter used.

LN131: Please provide the model number and apparatus manufacturer details.

LN130: Please provide the model number and manufacturer details of the Kjeldahl apparatus used.

LN141: Please provide the model number and manufacturer details of the Muffle furnace used.

Results:

LN156: Figure 1 should be placed after the relevant description/citation. So, please move down. Y axis: temperature: Please use Capital T: Temperature; humidity: Humidity

X-Axis: Better name as: Age (Weeks)

LN161: Better organize the content under subthemes: Growth performance for1-21 d; 22-42-d and 1-42-d for better clarity.

LN162: Have to state clearly that the test period is whether from day-1 or day-4.

LN162: All non-significant effects should be presented as (P>0.05).

LN165-LN167: Please revise as feed to gain ratio to sustain the uniformity. 

LN182: Figure 2: Daily weight gain and Daily weight gain: Units must be provided in parenthesis.

LN186: All non-significant differences must be supported by: (P>0.05). X-Axis: Better correct as: Age (Days)

LN190-LN193: Should not discuss numerical differences unless there is a statistical significance.

LN197: All non-significant differences must be supported by: (P>0.05).

LN218: Delete '': Please provide the staining details and magnification in parenthesis.

Discussion & Conclusion: No comments.

Reference List:

Complete. No missing references or citations.

Author Response

(The authors gave the same response as above.)

Reviewer 3 Report

Comments and Suggestions for Authors

Review (Animals), 13.11.2023

„The impact of different humidity levels on the production performance, slaughter performance, and meat quality of white Pekin ducks aged 4 to 42 days”

            The reviewed manuscript "The impact of different humidity levels on the production performance, slaughter performance, and meat quality of white Pekin ducks aged 4 to 42 days" is an interesting research work in the field of breeding white Pekin ducks. New duck breeding lines developed in recent years are characterized by high efficiency, but at the same time they are more sensitive to environmental influences. Therefore, it is important to conduct research on the impact of individual environmental factors on production results and the quality of duck meat. For this reason, I assess positively the choice of the research topic and its usefulness for science and the practice of breeding ducks for slaughter.

In the introduction to the manuscript, the authors introduced the reader well to the subject matter and correctly justified the validity of the research undertaken. This chapter ends with a well-formulated research objective. The material and methods are appropriately selected for the research and analyzes performed and are clearly described. I wonder whether the referenced standards (5009.5-2016; GB/T5009.4-2010) should not be listed with a bibliographic description in the reference list.

The results of the conducted tests are presented in two tables (table 2 and 3) and three figures (figure 2, 3 and 4), which is sufficient. In Table 2, the "Item" column should be corrected in accordance with the data in the other columns and the assigned units should be corrected. The "abdominal fat" position is repeated. The discussion of the results is concise and generally correct. However, the numbers of tables and figures discussed in the text should be cited. The slaughter performance chapter discusses data that do not differ significantly (abdominal fat ratio), but does not mention significant differences (liver weight and spleen index). The discussion is substantively correct and the literature used is well selected. Both the results and the discussion do not mention specific research results - which makes them somewhat general. The research conclusions are too laconic. They need to be reworded. All items from the reference list are carefully selected and used in the publication. The writing of references in Chinese should be changed (items 1, 14, 17, 32, 33, 34, 35). I suggest writing the authors' names in Latin alphabet and the titles of the works in English and in brackets (in Chinese). Editorial corrections are noted below.

p. 1, line 12:  Simple Summary: Humidity is a crucial environmental factor influencing duck growth. …

                      Humidity is one of the crucial environmental factor influencing duck growth.

p. 1, lines 29-31: Group C had significantly higher breast muscle weight, breast muscle ratio, liver weight, and

                                liver index than Groups B and…….              – see table 2

p. 2, line 87:  From days 1 to 14, the brooding temperature gradually decreased from 35°C to 26°C, and……

                                                        the rearing or breeding ?

p. 4, line 157:   Figure 1. This is a figure. Schemes follow the same formatting.

                                       ??? The title should match the content

p. 6, lines 199-203  When compared to Group A, Groups B and D show no significant differences in brightness (L* value), while Group C exhibits a significant decrease in brightness (L* value) (P<0.05), and there are no significant differences in brightness (L* value) between Groups B and D when compared to Group C.   - reword

p. 6, lines 207-208Group C exhibits significantly higher breast muscle weight and breast muscle percentage

                                      compared to other treatment groups (P<0.05).

                              repetition from the chapter “slaughter performance”

p. 6, line 208-209:  There are no significant differences in pH, 208 yellowness (b* value), fat, protein, ash content,

                                     and cooking loss.       – see table 3

p. 8, lines 265-266:  One of the primary objectives of animal husbandry is to provide high-quality protein sources

       for humans. One of the primary objectives of animal husbandry is to provide high-quality

       protein sources for humans.    – reduplication

The manuscript " The impact of different humidity levels on the production performance, slaughter performance, and meat quality of white Pekin ducks aged 4 to 42 days " is an interesting and substantively correct, valuable scientific work. After minor corrections, it may be published in Animals magazine.

Author Response

(The authors gave the same response as above.)

Round 2

Reviewer 1 Report

Comments and Suggestions for Authors

The authors considered the suggestions and made the pertinent changes to the manuscript.

Comments on the Quality of English Language

I suggest that the manuscript will be reviewed once again by an expert to ensure that it is published with the best possible quality.